# Partial Cystectomy for Muscle-Invasive Bladder Cancer

**DOI:** 10.3390/cancers17152562

**Published:** 2025-08-03

**Authors:** Peter S. Palencia, Nethusan Sivanesan, Syed Rahman, Fady Ghali, David Hesse, John Colberg, Ashwin Sridhar, John D. Kelly, Byron H. Lee, Ashish M. Kamat, Wei-Shen Tan

**Affiliations:** 1Department of Urology, Yale School of Medicine, New Haven, CT 06520, USA; 2Department of Urology, University College London Hospitals, London W1G 8PH, UK; 3Division of Surgery & Interventional Science, University College London, London W1W 7TY, UK; 4Department of Urology, University of Texas MD Anderson Cancer Center, Houston, TX 77030, USA

**Keywords:** muscle-invasive bladder cancer (MIBC), urachal adenocarcinoma, diverticular bladder tumors, Partial cystectomy

## Abstract

Surgical excision to remove part of the bladder (partial cystectomy) is an option for selected patients with muscle-invasive bladder cancer, urachal cancer and tumor within a pouch in the bladder (diverticulum). In this review, we discuss the current literature on partial cystectomy, indications, patient selection, initial workup, surgical approach and the role of systemic therapy. We also compare complications, cancer outcomes and quality of life between patients treated with partial cystectomy and complete bladder removal (radical cystectomy).

## 1. Introduction

Radical cystectomy is the standard of care for muscle-invasive bladder cancer (MIBC) and selected non-muscle-invasive bladder cancer (NMIBC). While radical cystectomy offers effective cancer control, it is associated with a >50% 90-day overall complication rate, 15–20% major complication rate and a 2% risk of mortality [1,2]. This results in some patients declining radical cystectomy or not being deemed fit enough for surgery due to co-morbidities. Additionally, radical cystectomy results in urinary and sexual-function adverse events and can have quality-of-life implications (QoL), resulting in poorer physical, cognitive, emotional, and social QoL domains [2].

Efforts to minimize patient morbidity have led to the development of bladder-sparing treatments while accepting a compromise of a higher recurrence risk. Conventionally, bladder preservation treatment is often delivered as chemoradiation therapy. Partial cystectomy is often forgotten as a treatment option in selected bladder cancer patients.

There is no definitive consensus on the use of partial cystectomy for bladder cancer. The American Urological Association (AUA) does not recommend partial cystectomy, and the European Association of Urology (EAU) does not provide specific guidance on its use [3,4]. The National Comprehensive Cancer Network (NCCN) considers partial cystectomy after neoadjuvant chemotherapy (NAC) a category 2A recommendation for highly selected cT2 bladder cancer patients with a solitary lesion in an appropriate location with the absence of carcinoma in situ (CIS) [5]. The International Bladder Cancer Group (IBCG) states that partial cystectomy can be discussed as an alternative to radical cystectomy or chemoradiation in carefully selected non-variant urothelial cancer histological subtype MIBC without CIS, where tumor is small and solitary and resection can be performed confidently with negative surgical margins [6]. It is estimated that only 5–10% of patients are suitable candidates for partial cystectomy [7].

In this review, we will synthesize the literature regarding patient selection, patient workup, surgical approach, the role of systemic therapies, and oncological outcomes following partial cystectomy.

## 2. Methods

This narrative review was performed using Pubmed using the following MeSH terms: “bladder cancer”, “muscle invasive bladder cancer”, “partial cystectomy”, “radical cystectomy”, “lymph node dissection”, “oncological outcomes”, “systemic therapy” and “minimal invasive surgery”.

### Inclusion and Exclusion Criteria

Patient selection for partial cystectomy is crucial to ensure oncological outcomes are not compromised. Patients with bladder diverticulum tumors and urachal adenocarcinoma are ideal candidates for partial cystectomy.

A retrospective study of eleven North American and European hospitals, the largest case–control study of bladder diverticulum tumors to date, compared oncological outcomes for 115 patients treated with partial cystectomy or radical cystectomy. The study reported a local recurrence rate of 18% for partial cystectomy and demonstrated no difference in overall survival and metastasis-free survival between the two procedures [8]. A smaller study of 36 patients from New York University similarly reported no difference in progression-free survival between radical cystectomy and partial cystectomy for patients with diverticular bladder diverticulum [9].

Data to support partial cystectomy for urachal adenocarcinoma is based on retrospective studies. A study by Mayo Clinic reported 46 patients undergoing partial cystectomy and 14 patients undergoing radical cystectomy. Mayo Clinic found that the local recurrence rate was 15% for partial cystectomy and that there was no difference in overall survival between patients undergoing partial and radical cystectomy [10]. Likewise, a large retrospective study consisting of 841 patients with urachal carcinoma utilizing the National Cancer Database found no difference in overall survival between partial cystectomy and radical cystectomy [11].

With regard to conventional urothelial bladder cancer, the NCCN recommends that patients with solitary cT2 in a suitable location, defined as a location suitable for segmental resection with adequate margins, are suitable candidates for partial cystectomy. It is also essential that the extension of resection to ensure negative margins does not significantly reduce bladder capacity, resulting in irritative lower urinary tract symptoms. The ideal location is a solitary tumor at the bladder dome <3 cm in diameter, which facilitates ease of resection and reconstruction. Trigonal or bladder neck tumors are relative contraindications due to difficulty in ensuring negative margins during excision as well as preserving bladder function. The requirement for ureteric reimplant is not an absolute contraindication. All patients should be evaluated for the absence of carcinoma in situ (CIS), which is a contraindication for partial cystectomy.

Initial studies that explored partial cystectomy for MIBC report 5-year recurrence-free survival (RFS) of between 39 and 67% [7,12]. These high recurrence rates are due to the inclusion of patients with multifocal disease and CIS. An analysis of 58 bladder cancer patients from Memorial Sloan Kettering Cancer Center (MSKCC) reported that patients with multifocal disease (56% vs. 8%) and CIS (33% vs. 9%) had a significantly higher risk of NMIBC recurrence [12]. Patients with multifocal disease (*p* < 0.001) and CIS (*p* = 0.027) were significantly more likely to develop local recurrence.

Studies performed by MD Anderson Cancer Center (MDACC) and the University of Pittsburg Medical Center (UPMC) applied more stringent criteria for MIBC patients undergoing partial cystectomy, including patients with solitary lesions amenable to a complete resection with clear surgical margins, and excluded patients with multifocal disease and CIS [7,13]. MDACC also excluded patients with tumors near the trigone or bladder neck to reduce the requirement for ureteral reimplantation. NMIBC recurrence rates following partial cystectomy were between 8 and 24% [7,13].

## 3. Workup

Perioperative workup for partial cystectomy would be similar to patients who are undergoing a radical cystectomy. All patients should have standard laboratory tests such as complete blood count, complete metabolic panel and urine culture tests. Assessing renal function is important to prioritize before using cisplatin-based neoadjuvant chemotherapy.

Cross-sectional imaging with CT of the chest, abdomen, and pelvis is recommended to ensure no evidence of metastasis. A urographic phase CT is useful to determine the path of the ureter and bladder diverticulum (where present) to determine the likelihood of requiring a ureteric reimplant. MRI of the pelvis may be helpful to better access local staging and surgical planning to ensure the absence of T3 disease. Similarly to how MIBC patients are managed prior to radical cystectomy, patients with suspected lymph node positive disease following perioperative chemotherapy are recommended to have a CT-guided biopsy. Where lymph node metastasis is histologically confirmed, further systemic therapy would be preferred before surgical consolidation. Where suspected, distant disease may be challenging to biopsy due to precarious anatomy, an FDG PET scan which has a sensitivity of 0.82 (95% CI: 0.72–0.89) and specificity of 0.89 (95% CI: 0.81–0.95) may be helpful [14].

Prior to partial cystectomy, performing a radical TURBT to grossly remove intraluminal tumor, which minimizes tumor spillage, is recommended. Prostatic urethral biopsies, as well as mapping bladder biopsies or blue light cystoscopy-guided biopsy, should be performed to rule out diffuse multifocal disease or CIS [6]. Blue light cystoscopy has been shown to improve the detection of CIS by 40.8% over white light cystoscopy [15]. Patients considering partial cystectomy should have good bladder function, tolerable lower urinary tract symptoms, and adequate bladder capacity after complete resection.

## 4. Role of Systemic Therapy

The principles of neoadjuvant systemic therapy for patients undergoing partial cystectomy are based on recommendations from radical cystectomy for MIBC, which has been shown to downstage tumors, treat micro metastases, reduce recurrence rates, and improve overall survival (OS) [16]. However, historical partial cystectomy series did not incorporate neoadjuvant systemic therapy [7,12,13]. A meta-analysis from 11 randomized control trials (RCTs) suggests that cisplatin-based neoadjuvant chemotherapy results in a 5% absolute overall survival advantage at 5 years [17]. A phase III RCT of MIBC patients randomized to neoadjuvant methotrexate, doxorubicin, vinblastine, and cisplatin (M-VAC) with radical cystectomy versus radical cystectomy alone concluded that neoadjuvant chemotherapy did not result in a delay to radical cystectomy, and there was no difference in health-related QoL outcomes following radical cystectomy [18].

Recent developments with novel systemic therapies have changed the landscape for neoadjuvant chemotherapy as the standard of care. NIAGARA, a phase III RCT, reported that the combination of gemcitabine-cisplatin with durvalumab resulted in an absolute OS advantage of 7% at 24 months compared to patients treated with gemcitabine-cisplatin neoadjuvant therapy [19]. EV-304 (NCT04700124), a phase III RCT evaluating perioperative enfortumab vedotin plus pembrolizumab therapy versus neoadjuvant chemotherapy for cisplatin-eligible MIBC, has completed patient recruiting, with results eagerly awaited given impressive results of enfortumab vedotin plus pembrolizumab therapy as a first-line treatment in the metastatic setting [20].

The availability of more efficacious systemic therapy may increase the number of patients suitable for bladder preservation treatment. Gemcitabine-cisplatin plus durvalumab and enfortumab vedotin plus pembrolizumab had a complete response rate of 33.8% and 60% at the time of radical cystectomy, suggesting consolidating patients with a partial cystectomy where anatomy is favorable is an option [19,21].

Limitations in imaging and tissue sampling from TURBT make it challenging to decipher which patients are truly T0 following neoadjuvant chemotherapy. In a retrospective series of 63 MIBC patients who had T0 at repeat TURBT and CT confirming downstaging of local bladder tumor following neoadjuvant chemotherapy, and who declined radical cystectomy, 54% of patients were alive with an intact bladder at a median follow-up time of 108 months [22]. A total of 38% of patients, however, subsequently developed MIBC with a median time to muscle invasion of 16 months, and only 25% of patients who developed MIBC recurrence were alive [22]. This highlights the importance of consolidation local therapy either in the form of surgery (partial or radical cystectomy) or chemoradiation therapy, even in patients who achieved a good response to systemic therapy.

Predictive biomarkers to neoadjuvant chemotherapy, such as ERCC2, have been validated using retrospective clinical trial cohorts to predict response to cisplatin-based therapy (OR: 8.3, 95% CI: 1.4–91.4, *p* = 0.01) [23]. External validation of three genes—ATM, ERCC2, and RB1—utilizing the SWOG S1314 patient cohort who received cisplatin-based neoadjuvant chemotherapy suggests that patients that harbor two or more mutations have a 67% probability of achieving T0 at radical cystectomy, compared to 14% for patients with no mutations [24]. Nevertheless, a prospective clinical trial has not validated these biomarkers externally. A phase II trial utilizing an algorithm-generated gene expression model, COXEN, failed to predict response to cisplatin-based neoadjuvant chemotherapy [25]. Hence, currently, there remains no validated tool to accurately predict response to neoadjuvant chemotherapy.

## 5. Advantages of Surgical Bladder Preservation Therapy

It is estimated that 5–10% of patients with MIBC are candidates for partial cystectomy with chemoradiation much more widely used as a bladder preservation option [7]. Acknowledging limitations in case selection, a multicenter, retrospective, propensity score-matched analysis of MIBC patients treated with trimodal therapy versus radical cystectomy, no difference was reported at 5-year follow up for metastasis-free survival (74% vs. 74%) and cancer-specific survival (83% vs. 8%) [26].

Surgical bladder preservation therapy holds several advantages over trimodal therapy in patients who fulfill the inclusion criteria. Achieving full-thickness definite pathology and determining margin status provides useful prognostic information. In an analysis of 548 patients with MIBC, >60% of patients who had pT0 at repeat TURBT had residual cancer at radical cystectomy [27]. This highlights TURBT as a poor tool for local restaging. Partial cystectomy would also avoid complications attributed to radiation therapy, such as hemorrhagic cystitis, rectal bleeding, irritative lower urinary tract symptoms, and secondary pelvic cancers [28]. The NCCN and IBCG have both recommended pelvic lymph node dissection when partial cystectomy is performed [5,6]. Radiating the pelvic nodes is typically not performed as part of trimodal therapy [29]. In patients with local recurrence of MIBC following partial cystectomy, they would remain candidates for trimodal therapy or radical cystectomy without the additional complicity of desmoplastic reaction attributed to radiotherapy-treated tissues, which can increase the risk of surgical complications such as rectal injury following consolidative radical cystectomy [30].

## 6. Surgical Approach

Partial cystectomy can be performed either by conventional open surgery or a minimally invasive approach, depending on surgeon preference.

For an open approach, the patient is positioned in lithotomy with Allen stirrups and slight Trendelenburg. A Foley catheter is inserted after the patient has been cleaned and draped. Depending on the location of the planned partial cystectomy, an extraperitoneal or transperitoneal approach can be performed through a lower midline incision. A pelvic lymph node dissection can be performed prior to or after the partial cystectomy.

A combination of an endoscopic view using a flexible cystoscopy and an open view is helpful to localize the area for a partial cystectomy. The bladder can be filled via the flexible cystoscope to allow the bladder to distend. The light from the flexible cystoscopy and tactile feeling can serve as a guide to determine precisely where bladder resection should be performed with a 1–2 cm margin. Stay sutures at the superior, inferior, right, and left lateral would be useful to outline the exact full thickness excision, including perivesical fat and peritoneum, should be performed. Before excision, the bladder is drained completely.

In patients with a posterior or lateral tumor, bladder mobilization, division of the vas deferens and ipsilateral bladder pedicle with Ligasure (Medtronic, Dublin, Ireland) will be required. A ureteric reimplant may be necessary where the tumor is located over or adjacent to the ureteric orifice. A frozen section is helpful to ensure margins are free from tumor. The bladder is then closed in two layers and a pelvic drain placed.

For a minimally invasive approach, the patient is positioned in a similar fashion with Allen stirrups with a steep Trendelenburg to displace the bowel from the pelvis. The robotic approach utilizes a six-port transperitoneal approach, with a similar configuration to a robotic prostatectomy, with all ports positioned 2 cm more cranially. This comprises four 8 mm robotic ports, a 5mm Airseal port (ConMed, Largo, FL) for the sucker/irrigator, and a 12 mm assistant port. The Pneumoperitoneum is set at 10 mm Hg. For posterior tumors, we could advocate preserving the space of Retzius and leaving the bladder supported at the anterior abdominal wall to facilitate posterior dissection. For lateral tumors, the peritoneum is incised lateral to the medial umbilical ligament, and the lateral bladder wall and ureter may require mobilization (Figure 1). A similar approach to open surgery can be adapted to the robotic approach, utilizing a flexible cystoscopy and stay sutures to outline the circumferential margin of full-thickness excision.

For diverticular tumors, the diverticulum is mobilized until the diverticulum neck is well defined (Figure 2). The bladder is incised at 12 o clock and the neck of the bladder diverticulum can be sutured closed from the inside to stop tumor spillage (Figure 3) and the entire bladder diverticulum mobilized off the bladder (Figure 4). In cases where a ureteric reimplant is required, a psoas hitch may be helpful to minimize tension at the ureteric vesical anastomosis (Figure 5).

The robotic approach provides several potential advantages, including reduced recovery times and length of stay, and possibly lower complication rates. However, hypothetical concerns about tumor spillage or aerosolization regarding tumor seedings should be minimized by ensuring the bladder is drained prior to transecting the detrusor muscle. After entering the bladder, the bladder mucosa is excised circumferentially with a 1–2 cm margin without energy and the specimen is directly placed in an endocatch bag, allowing for immediate containment of the excised specimen. The use of stay sutures prior to incision of the bladder may aid the provision of tension during partial cystectomy excision.

## 7. Perioperative and Functional Outcomes of Partial Cystectomy

Retrospective studies evaluating MIBC patients undergoing partial cystectomy have reported an acceptable major complication rate of 4–7% at 30 days and 4–11% at 90 days [31,32]. At 90 days, major complications reported include hydronephrosis, lymphocele, infected lymphocele, fascial dehiscence, urosepsis, urinary leak, urinary tract infection, and acute respiratory distress syndrome [32]. The mortality rate associated with PC was low, with 0% at 30 days and 0–2% at 90 days, in contrast to radical cystectomy [31,32].

Health-related quality of life (HR-QoL) was assessed using the European Organization for Research and Treatment of Cancer Quality of Life Questionnaire-Core 30 (EORTC QLQ-C30). This measure is based on a scale of 0–100, with higher scores indicating better quality of life for functional and overall scales, while for symptom and single-item scores, higher scores correspond to reduced QoL. Postoperatively, the median (IQR) HR-QoL score for the partial cystectomy was better compared to radical cystectomy [79.2 (52.1–97.9) vs. 66.7 (50.0–83.3)] [31]. Patients who underwent partial cystectomy also demonstrated better outcomes in multi-item function scores, including physical, cognitive, and social functioning, than those who underwent radical cystectomy. Additionally, they reported lower symptom burden in fatigue, nausea/vomiting, dyspnea, appetite loss, constipation, and diarrhea.

A retrospective study by Leveridge et al. comparing partial cystectomy to radical cystectomy found a significantly shorter length of hospital stay in the partial cystectomy treated patients (8 days vs. 11 days; *p* < 0.001) [33]. Readmission rates were comparable between the two groups.

Overall, retrospective studies indicate that partial cystectomy demonstrates promise in terms of perioperative and functional outcomes. However, given the limited number of studies, further research is warranted. Specifically, randomized controlled trials are needed to directly compare the functional and perioperative outcomes of partial cystectomy with radical cystectomy.

## 8. Lymph Node Dissection

The current NCCN and IBCG guidelines recommend bilateral pelvic lymphadenectomy for patients undergoing partial cystectomy [5,6]. This recommendation is based on studies looking at radical cystectomy survival outcomes showing lymph node dissection as an important prognostic tool. However, whether lymph node dissection is curative remains debatable. One such study is the Herr study, which suggested that removal of >10 nodes was associated with a better post-cystectomy survival outcome [34].

A recent multicenter randomized trial of patients with cT2-4a N0-2 bladder cancer randomized to standard or extended lymphadenectomy reported that there was no difference in disease-free survival or overall survival [35]. Moreover, the study also showed extended lymphadenectomy being associated with more significant morbidity and higher peri-operative 90-day mortality relative to standard lymphadenectomy. The German LEA randomized control trial previously reported no oncological advantage in performing extended lymph node dissection relative to limited dissection in MIBC patients who did not receive neoadjuvant chemotherapy [36].

Retrospective analysis of 1647 patients with NMIBC undergoing radical cystectomy suggests that increased lymph node yield was associated with improvement in local pelvic recurrence-free survival, cancer-specific survival and OS [37]. However, such analysis may be confounded by the Will Rogers phenomenon because a higher lymph node yield selects for a ‘cleaner’ cohort of node negative patients, introducing selection bias.

In summary, when partial cystectomy is performed, a pelvic lymph node dissection should be performed as part of the standard of care. However, in NMIBC patients with significant comorbidities, omitting lymphadenectomy may be considered to minimize operative time and reduce potential surgical risks, acknowledging the low therapeutic benefits of lymph node dissection due to the low incidence of lymph node metastasis. Further studies with evidence from randomized controlled trials are needed to ensure whether the omission of lymphadenectomy is acceptable for patients with T1 bladder cancer.

## 9. Oncological Outcomes

Several retrospective studies have demonstrated acceptable cancer control following partial cystectomy for MIBC (Table 1). MSKCC conducted one of the first retrospective studies, which consisted of 58 patients with primary non-urachal bladder tumor (79% urothelial) and 71% had ≥cT2 bladder cancer [12]. The bladder dome (41%) was the most common site of cancer followed by the lateral wall (19%), anterior wall (14%) and bladder diverticulum (12%). Of note, 26% of patients had CIS, and 15% had multifocal disease. At a median follow-up of 31 months, 69% of patients remained recurrence free. Bladder recurrence was noted in 19% of patients, of which 12% had NMIBC recurrence. Five-year RFS and overall survival (OS) were 55% and 69%, respectively.

Other single intuitional studies performed by MDACC (*n* = 37) and UPMC (*n* = 25) supported partial cystectomy as a viable treatment for muscle-invasive bladder cancer [7,13]. Unlike the MSKCC study, these studies only included unifocal disease without CIS that could be resected with clear surgical margins. Bladder recurrence was reported in 24–35% of patients, with NMIBC recurrence accounting for 8–24% of patients, with a median follow up of 18–51 months. Between 24 and 40% of patients received adjuvant chemotherapy postoperatively. Five-year RFS and OS were 39–62% and 67–70%, respectively.

Mayo Clinic performed a propensity score-matching analysis in which 253 MIBC patients were matched (1—partial cystectomy: 2—radical cystectomy ratio) on age, gender, pathological stage, and receipt of neoadjuvant chemotherapy [38]. Patients undergoing partial cystectomy followed a strict inclusion criterion, only including those with a solitary lesion without associated CIS, and did not require ureteral reimplantation. At a mean follow-up of 74 months, 81% of patients retained their bladder. No difference in 5-year cancer-specific survival (68% vs. 72%, *p* = 0.7), metastasis-free survival (72% vs. 75%, *p* = 0.6), and OS (55% vs. 50%, *p* = 0.4) was reported following treatment with partial and radical cystectomy, respectively. A more recent case–control study which analyzed 102 MIBC patients treated with partial (*n* = 32) or radical *n* = 70) cystectomy collaborates these findings and reported no significant difference in 5-year OS (45% vs. 53%), RFS (55% vs. 62%), and cancer-specific survival (62% vs. 68%) between the partial and radical cystectomy, respectively [39].

Large retrospective database studies have compared oncologic outcomes between radical and partial cystectomy. Capitano et al. performed a propensity score-matching analysis with 1573 partial and 5670 radical cystectomy patients (1:4 ratio) utilizing the Surveillance, Epidemiology, and End Results (SEER) database [40]. Patient cohorts were matched for age, race, tumor grade, tumor stage, nodal stage, year of surgery, and lymph node yield. There was no demonstrated difference in 5-year overall survival (57% vs. 55%, *p* = 0.3) and cancer-specific survival (70% vs. 69%, *p* = 0.5) between the partial and radical cystectomy cohorts, respectively.

Chung et al. utilized the National Cancer Database (NCDB) to compare overall survival of 1457 partial cystectomy patients against 21,067 radical cystectomy patients [41]. Their primary analysis suggests that patients treated with radical cystectomy had better overall survival compared to partial cystectomy following Cox regression analysis (HR: 0.88, 95% CI: 0.80–0.95, *p* = 0.002). However, when patients with cT2, no-concurrent CIS with tumors <5 cm were selected, there was no difference in OS between treatment arms (HR: 1.02, 95% CI: 0.9–1.2, *p* = 0.7).

Kijima et al. reported outcomes of 154 patients with MIBC (<treated with TURBT followed by chemoradiation [42]. Patients with a complete response rate determined by absence of MIBC at restaging TURBT, negative MRI and urine cytology at 4–6 weeks after chemoradiation received pelvic lymph node dissection with patients with residual disease offered radical cystectomy. A total of 125 (81%) patients achieved a complete response after chemoradiation; 107 patients proceeded with a partial cystectomy, which confirmed residual tumor in 11 patients (10%), of which three patients had pT2 and six patients had pT3 cancer. Two patients had lymph node positive cancer. This highlights the limitations of TURBT and imaging as a staging tool following chemoradiation. Reported MIBC RFS, CSS and OS rates were 97%, 93% and 91%, respectively.

There remains a lack of high-quality data to support partial cystectomy. Discrepancies between retrospective data between studies comparing PC with radical cystectomy in retrospective fashion may in fact be impacted by inherent selection biases reflected in the incorporated studies. For instance, some of the studies that demonstrated that high RFS rates for patients with PC may be reflective of patients with multifocal disease and CIS being incorporated (12). Cases typically selected for partial cystectomy would be of ‘lower risk’ given that tumors are solitary and small enough that excision would allow for negative margins. Nevertheless, patients treated with partial cystectomy may have more comorbidities and might have been deemed too high risk for a radical cystectomy.

## 10. Surveillance and Follow-Up

The IBCG and NCCN surveillance guidance are similar. Following partial cystectomy, surveillance cystoscopy and urine cytology are recommended every 3 months for the first 24 months followed by every 6 months for year 3 and 4 and then yearly for life, which is consistent with high-risk NMIBC guidance [5,6,43]. Cross-sectional imaging of the chest, abdomen and pelvis is recommended every 3–6 months for the first 2–3 years and then annually for at least 5 years [5,6].

## 11. Conclusions

In conclusion, radical cystectomy is the standard of care for MIBC. However, partial cystectomy is not considered a standard option in MIBC. Acknowledging the limitations of retrospective case series or case–control studies, which are lower-quality evidence, it may be an alternative to radical cystectomy in carefully selected patients, such as those with small solitary cT2 disease amenable to resection with adequate margins that do not exhibit CIS or histological subtype (excluding pure adenocarcinoma of the urachus) and after adequate consultation about the risks versus benefits of this approach. Partial cystectomy is a safe and acceptable treatment option for patients with urachal adenocarcinoma or diverticular tumors. Mapping bladder biopsies or blue light cystoscopy and prostatic urethral biopsy is recommended to rule out CIS.

The ability for bladder preservation which allows for subsequent salvage radical therapy, lower surgical morbidity, a shorter length of stay and an earlier return to normality are advantages of partial cystectomy. Local bladder surveillance is crucial due to the risk of both MIBC and NMIBC recurrence, which should be managed in accordance with established right risk NMIBC guidelines and to allow prompt salvage radical cystectomy or radical radiotherapy in patients with MIBC recurrence. Neoadjuvant systemic therapy and pelvic lymph node dissection is recommended in MIBC patients undergoing partial cystectomy. Follow-up after PC for MIBC should be patient-specific and include lifelong annual cystoscopy.

## Figures and Tables

**Figure 1 cancers-17-02562-f001:**
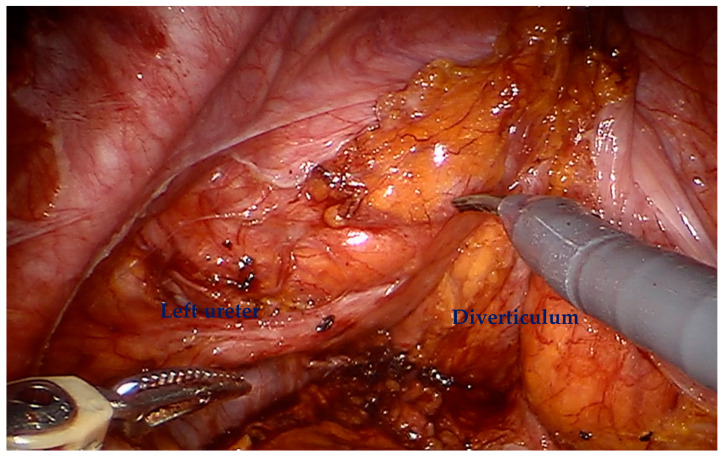
The peritoneum is incised, and the ureter is mobilized off the bladder diverticulum.

**Figure 2 cancers-17-02562-f002:**
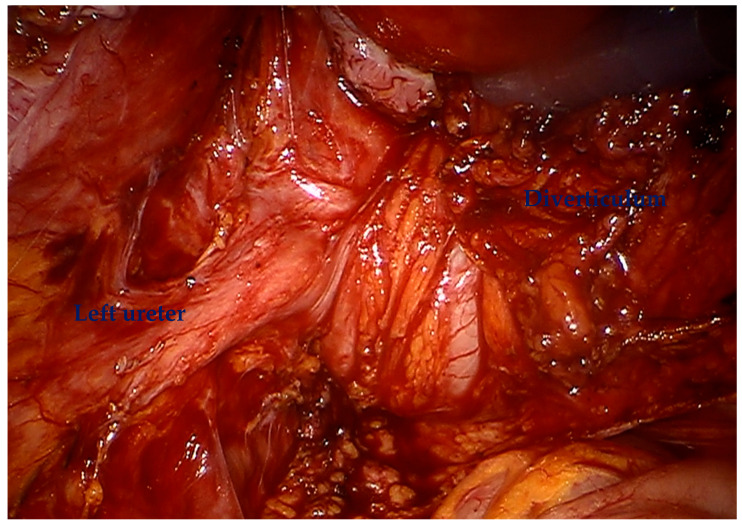
The bladder diverticulum is mobilized until the diverticulum neck is well defined.

**Figure 3 cancers-17-02562-f003:**
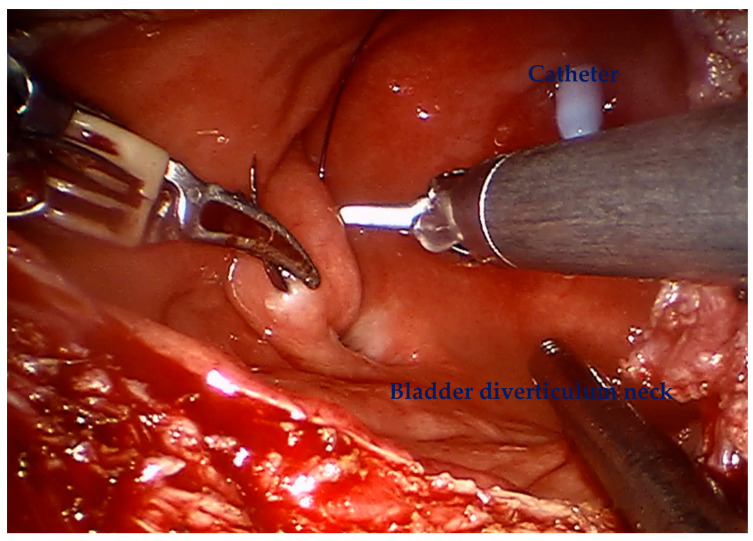
At 12 O′clock away from the bladder diverticulum, the bladder is opened. The bladder diverticulum intravesically is closed to prevent tumor spillage.

**Figure 4 cancers-17-02562-f004:**
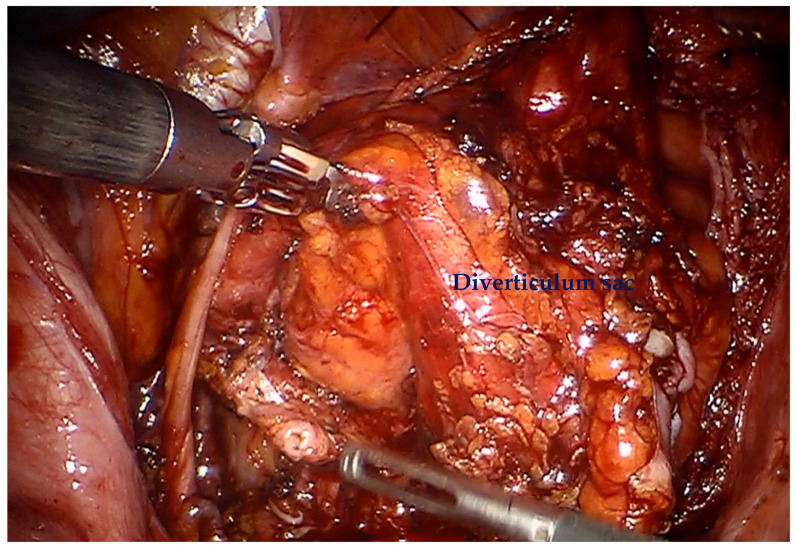
Complete mobilization of the entire bladder diverticulum.

**Figure 5 cancers-17-02562-f005:**
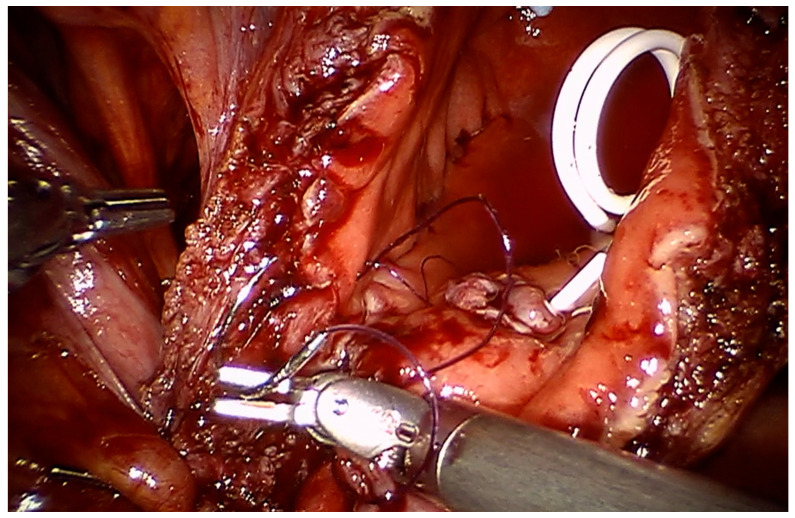
Psoas hitch with ureteric reimplant and stent insertion is performed, and bladder is closed.

**Table 1 cancers-17-02562-t001:** Summary of oncological outcomes for partial cystectomy.

Study	No. Patients	Inclusion Criteria	Exclusion Criteria	5-year OS	5-year RFS	5-year CSS	5-year MFS	Mean-Follow Up (Months)
Holzbeierlein et al. [12]	58 PC	Primary bladder tumors of nonurachal origin, including multifocal disease and/or CIS	Urachal tumors	69%	67%	***	***	33
Kassouf et al. [7]	37 PC	Solitary bladder tumor without CIS, ≥2 cm negative surgical margins, requires ureteral reimplantation if needed	Carcinoma in situ (CIS), multifocal tumors, inadequate margins	67%	39%	87%	***	73
Smaldone et al. [13]	25 PC	Solitary urothelial tumor without CIS, negative surgical margin; protocol includes adjuvant RT (5 doses of 25 Gy)	CIS, positive surgical margins	70%	62%	84%	***	45
Knoedler et al. [38]	167 RC; 86 PC	T1–T4, N0 or N+, M0 bladder cancer; PC vs. RC (1:2 matched case–control)	Metastatic disease (M1)	PC: 55% RC: 50%	***	PC: 68% RC: 72%	PC: 72% RC: 75%	PC: 74 (median), RC:74 (median)
Zhang et al. [39]	70 RC; 32 PC	T2–T3, N0, M0 urothelial carcinoma; PC vs. RC (case–control)	Metastatic disease (M1)	PC: 45% RC: 53%	PC: 55% RC: 62%	PC: 62% RC: 68%	***	PC: 32 (median); RC: 23 (median)
Capitanio et al. [40]	5670 RC; 1570 PC	T1–T4, N0–2, M0 urothelial carcinoma; PC vs. RC (1:4 matched)	Metastatic disease (M1)	PC: 57% RC: 55%	***	PC: 70% RC: 69%	***	PC: 64; RC: 77
Chung et al. [41]	21067 RC; 1457 PC	T2–T4, N0 or N+, M0; PC vs. RC (case–control)	Metastatic disease (M1)	PC: 50% RC: 52%	***	***	***	PC: 34 (median), RC:34 (median)
Kijima et al. [42]	107 PC	Muscle-invasive bladder cancer (MIBC); chemoradiation with bladder preservation	Non-MIBC, poor performance status	91%	97 (MIBC)	93%	***	48

PC—partial cystectomy, RC—radical cystectomy, OS—overall survival, RFS—recurrence-free survival, CSS—cancer-specific survival, MFS—metastasis free survival; ***—not reported.

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
