# Peer review of "Partial Cystectomy for Muscle-Invasive Bladder Cancer"

_cancers, 2025, doi:10.3390/cancers17152562_

Round 1
Reviewer 1 Report
Comments and Suggestions for Authors
The authors reviewed the literature on partial cystectomy for muscle-invasive bladder cancer. The article is well-organized and informative for readers. The reviewer would like to offer a few suggestions to further improve the manuscript.
Major Comments:
- As described in the introduction, current guidelines do not actively recommend partial cystectomy as a bladder-preserving option for bladder cancer. The authors may wish to more explicitly clarify the rationale for focusing on partial cystectomy and to highlight its potential strength as an alternative strategy for bladder preservation.
- Recent clinical trials have explored several novel protocols, particularly those incorporating immune checkpoint inhibitors, which may become mainstream in bladder-preserving strategies for muscle-invasive bladder cancer in the near future. From this perspective, the authors are encouraged to further discuss the potential future role of partial cystectomy as an alternative within this evolving treatment landscape.
Minor Comment:
- The following study reported that partial cystectomy may offer a unique benefit as a consolidative treatment following induction chemoradiotherapy:
Kijima T, et al.Selective tetramodal bladder-preservation therapy, incorporating induction chemoradiotherapy and consolidative partial cystectomy with pelvic lymph node dissection for muscle-invasive bladder cancer: oncological and functional outcomes of 107 patients. BJU Int. 2019; 124: 242–250.
The authors may consider citing this reference and discussing its relevance in the context of the current review.
Author Response
Comment 1:
The following study reported that partial cystectomy may provide a unique benefit as a consolidative treatment following induction chemoradiotherapy:
Kijima T, et al. Selective tetramodal bladder-preservation therapy, incorporating induction chemoradiotherapy and consolidative partial cystectomy with pelvic lymph node dissection for muscle-invasive bladder cancer: oncological and functional outcomes of 107 patients. BJU Int. 2019; 124: 242–250. The authors may consider citing this reference and discussing its relevance in the context of the current review.
Response 1:
We thank the reviewer for pointing out this manuscript. We have added it in the discussion regarding oncological outcomes and added in a paragraph to discuss this manuscript in lines 358-367. We have also added it to Table 1.
Reviewer 2 Report
Comments and Suggestions for Authors
This narrative review on partial cystectomy (PC) for muscle-invasive bladder cancer (MIBC) is thorough, well-written, and offers a synthesis of the current evidence. The authors successfully cover key aspects such as patient selection, surgical techniques, systemic therapies, and oncologic outcomes, complemented by excellent intraoperative images that enhance the understanding of surgical details. The manuscript flows well and is logically organized. Nonetheless, there are important areas that merit further development to maximize the review’s clinical impact and academic rigor.
Although the article is a narrative review, it lacks even a brief description of how the literature was sourced. Including a short methods section that details which databases were searched, keywords used, and criteria for inclusion would significantly improve the transparency and reproducibility of the work. Additionally, while the authors rightly highlight that much of the evidence comes from retrospective and single-institution series—given the rarity and complexity of PC—the manuscript would benefit from a more explicit summary of the overall quality and hierarchy of the evidence supporting PC in MIBC.
A notable point for clarification involves patient selection criteria. The manuscript references differing inclusion criteria across leading institutions such as MSKCC, MDACC, and UPMC. These variations are critical, as they influence both patient outcomes and the generalizability of study findings. A comparative table summarizing inclusion and exclusion criteria, alongside outcomes from key studies, would provide readers with a clearer framework for interpreting results and applying them in clinical practice.
Regarding oncologic outcomes, while the review provides useful data comparing PC with radical cystectomy, it stops short of analyzing in depth why some studies report no significant difference in overall survival while others do. A deeper exploration of potential selection biases, confounding factors, and patient characteristics—such as the likelihood that healthier or lower-risk patients are chosen for PC—would add valuable nuance to the discussion and help readers interpret conflicting data.
Overall, the manuscript demonstrates significant strengths, including a comprehensive examination of the topic, clear explanation of surgical procedures, discussion of rare histologic variants such as urachal carcinoma, and inclusion of recent developments in systemic therapy. The practical, guideline-based recommendations for post-operative surveillance are another valuable feature.
This is an important and timely review on a complex surgical option for MIBC. However, to elevate the manuscript further, the authors should provide greater transparency regarding their literature search, explicitly discuss the quality of supporting evidence, analyze the variability in selection criteria across institutions, and delve deeper into the interpretation of comparative oncologic outcomes.
Author Response
Reviewer 2
This narrative review on partial cystectomy (PC) for muscle-invasive bladder cancer (MIBC) is thorough, well-written, and offers a synthesis of the current evidence. The authors successfully cover key aspects such as patient selection, surgical techniques, systemic therapies, and oncologic outcomes, complemented by excellent intraoperative images that enhance the understanding of surgical details. The manuscript flows well and is logically organized. Nonetheless, there are important areas that merit further development to maximize the review’s clinical impact and academic rigor.
Comment 1:
Although the article is a narrative review, it lacks even a brief description of how the literature was sourced. Including a short methods section that details which databases were searched, keywords used, and criteria for inclusion would significantly improve the transparency and reproducibility of the work. Additionally, while the authors rightly highlight that much of the evidence comes from retrospective and single-institution series—given the rarity and complexity of PC—the manuscript would benefit from a more explicit summary of the overall quality and hierarchy of the evidence supporting PC in MIBC.
Response 1:
We thank the reviewer for his/ her comments.
We have added a short methods section in lines 67- 71 with the following:
Methods
This narrative review was performed using Pubmed using the following MeSH terms “bladder cancer”, “muscle invasive bladder cancer”, “partial cystectomy”, “radical cystectomy”, “lymph node dissection”, “oncological outcomes”, “systemic therapy” and “minimal invasive surgery”.
As pointed out by the reviewer, studies discussed were all retrospective case series or case control studies hence the reports represents poor quality evidence. We have added the following in the conclusions (lines 391-396):
‘Acknowledging limitations of retrospective case series or case control studies which are lower quality evidence, it may be alternative to radical cystectomy in carefully selected patients such as those with small solitary cT2 disease amenable to resection with adequate margins that do not exhibit CIS or histological subtype (excluding pure adenocarcinoma of the urachus) after adequate consultation about risks versus benefits of this approach.’
Comment 2:
A notable point for clarification involves patient selection criteria. The manuscript references differing inclusion criteria across leading institutions such as MSKCC, MDACC, and UPMC. These variations are critical, as they influence both patient outcomes and the generalizability of study findings. A comparative table summarizing inclusion and exclusion criteria, alongside outcomes from key studies, would provide readers with a clearer framework for interpreting results and applying them in clinical practice.
Response 2:
We have added study inclusion and exclusion criteria in Table 1.
Comment 3:
Regarding oncologic outcomes, while the review provides useful data comparing PC with radical cystectomy, it stops short of analyzing in depth why some studies report no significant difference in overall survival while others do. A deeper exploration of potential selection biases, confounding factors, and patient characteristics—such as the likelihood that healthier or lower-risk patients are chosen for PC—would add valuable nuance to the discussion and help readers interpret conflicting data.
Response 3:
We have added a paragraph in lines 373- 381 to discuss this:
‘There remains a lack of high-quality data to support partial cystectomy. Discrepancies between retrospective data between studies comparing PC with radical cystectomy in retrospective fashion may in fact be impacted by inherent selection biases reflected in the incorporated studies. For instance, some of the studies that demonstrated that high RFS rates for patients with PC may be reflective of patients with multifocal disease and CIS being incorporated (12). Cases typically selected for partial cystectomy would be of ‘lower risk’ given that tumors are solitary and small enough that excision would allow for nega-tive margins, Nevertheless, patients treated with partial cystectomy may be more comorbid patients who might have been deemed too high risk for a radical cystectomy.’
Overall, the manuscript demonstrates significant strengths, including a comprehensive examination of the topic, clear explanation of surgical procedures, discussion of rare histologic variants such as urachal carcinoma, and inclusion of recent developments in systemic therapy. The practical, guideline-based recommendations for post-operative surveillance are another valuable feature.
This is an important and timely review on a complex surgical option for MIBC. However, to elevate the manuscript further, the authors should provide greater transparency regarding their literature search, explicitly discuss the quality of supporting evidence, analyze the variability in selection criteria across institutions, and delve deeper into the interpretation of comparative oncologic outcomes.
Reviewer 3 Report
Comments and Suggestions for Authors
Comments for the article titled, “Partial Cystectomy for Muscle Invasive Bladder Cancer” are mentioned below
- The first sentence of the abstract section, line Nos 12-13, “Partial cystectomy is a surgical bladder-sparing option for selected patients with muscle invasive bladder cancer (MIBC), urachal adenocarcinoma and certain diverticular bladder tumors and remains underutilized.” is difficult to understand. Revise and rewrite for clarity.
- The abstract section does not adhere to the IMRaD format.
- The introduction section is informative and supported by appropriate references.
- What is the source of these figures included in the manuscript? Since this is a review article, are the figures the authors' own research contributions or taken from some other source?
- I cannot locate the in-text mention of Table 1.
- Is the manuscript a literature review or a systematic review? If this is a systematic review, include a PRISMA flow chart. The article type seems confusing.
- Mention the strengths and weaknesses of this manuscript.
Author Response
Comment 1:
- The first sentence of the abstract section, line Nos 12-13, “Partial cystectomy is a surgical bladder-sparing option for selected patients with muscle invasive bladder cancer (MIBC), urachal adenocarcinoma and certain diverticular bladder tumors and remains underutilized.” is difficult to understand. Revise and rewrite for clarity.
Response 1:
We have shorted the first sentence of the abstract to the following:
‘Partial cystectomy is a surgical bladder-sparing option for selected patients with muscle-invasive bladder cancer (MIBC), urachal adenocarcinoma and diverticular bladder tumors.’
Comment 2:
- The abstract section does not adhere to the IMRaD format.
Response 2:
The abstract has been rewritten and is now a total of 212 words.
Comment 3:
- The introduction section is informative and supported by appropriate references.
- What is the source of these figures included in the manuscript? Since this is a review article, are the figures the authors' own research contributions or taken from some other source?
Response 3:
The figures are intraoperative screenshots of the authors cases.
Comment 4:
- I cannot locate the in-text mention of Table 1.
Response 4:
We apologize for this. This has now been added to line 321.
Comment 5:
- Is the manuscript a literature review or a systematic review? If this is a systematic review, include a PRISMA flow chart. The article type seems confusing.
- Mention the strengths and weaknesses of this manuscript.
Response 5:
No this is not a systematic review and represents a narrative review. We have now added a methods section as recommended by reviewer 1 which states this (lines 66-70):
- Methods
This narrative review was performed using Pubmed using the following MeSH terms “bladder cancer”, “muscle invasive bladder cancer”, “partial cystectomy”, “radical cys-tectomy”, “lymph node dissection”, “oncological outcomes”, “systemic therapy” and “minimal invasive surgery”.
We have added the following in lines 391-396 to acknowledge limitations of the manuscript:
‘ Acknowledging limitations of retrospective case series or case control studies which are lower quality evidence, it may be alternative to radical cystectomy in carefully selected patients such as those with small solitary cT2 disease amenable to resection with adequate margins that do not exhibit CIS or histological subtype (excluding pure adenocarcinoma of the urachus) after adequate consultation about risks versus benefits of this approach.’